# A Quickly Deployed and UAS-Based Logistics Network for Delivery of Critical Medical Goods during Healthcare System Stress Periods: A Real Use Case in Valencia (Spain)

**Israel Quintanilla García *** , **Norberto Vera Vélez** , **Pablo Alcaraz Martínez** , **Jordi Vidal Ull**
and **Beatriz Fernández Gallo**

Department of Cartographic Engineering, Geodesy and Photogrammetry, Universitat Politècnica de València, Camino de Vera, s/n, Edificio 7i, E46022 Valencia, Spain; norvevle@doctor.upv.es (N.V.V.); pabalma4@upv.es (P.A.M.); jorviul@fiv.upv.es (J.V.U.); beaferga@topo.upv.es (B.F.G.)
* Correspondence: iquinta@cgf.upv.es

**Abstract:** On the one hand, Unmanned Aircraft Systems (UASs) have experienced great applicability surge in the recent years, arising as a promising technology with a wide field of use. On the other hand, healthcare, a critical system in modern society, is subject to a heavy and unexpected pressure in the case of situations such as the COVID-19 pandemic. This article aims to leverage the flexibility of UASs as complementary support for healthcare logistic systems when under high-stress conditions, via quick deployment of an air delivery network. We have defined a logistics network model and created three scenarios based on the model and current needs in Valencia (Spain). Flight tests have been performed in these scenarios, which include urban areas and controlled airspace. Operations complied with requirements derived from the application of Specific Operations Risk Assessment (SORA) methodology, recently adopted by the European Aviation Safety Agency (EASA). Flights were successful, being able to swiftly deliver medical goods without requiring any dedicated infrastructure. However, a moderate number of contingencies took place during the tests, mainly related to control link quality and Air Traffic Management (ATM) integration, forcing the use of dedicated procedures to cope with them. Although additional development is required to ensure the safety of large-scale automated operations, the use of UASs as part of logistic networks is a feasible means to support existing structures, especially in situations in dire need.

**Keywords:** UAS; urban; BVLOS; drone; medicines drone delivery ; healthcare logistics; SORA

## 1. Introduction

Colloquially known as drones, these versatile tools have risen as a solution in many different fields, including aerial imagery, agriculture, mapping, inspections, or search and rescue, among others. Although a great tool from a theoretical point of view, and with a huge predicted market [1], practical deployment of drones faces several barriers that need to be tackled, especially regarding regulation and public perception [2]. In most countries, drone operations are restricted to visual line of sight (VLOS) conditions (i.e., the pilot must keep the drone inside their field of vision) and must be performed in rural areas in uncontrolled airspace, in order not to pose a risk to people or other aircraft. Indeed, drones are considered aircraft as per the definition of the International Civil Aviation Organization (ICAO) [3], and drones are subject to compliance with aviation standards and procedures. Another critical barrier is the integration of drones in non-segregated airspace [4]. Drone regulation in each country is largely independent nowadays, leading to a low level of harmonization among countries.

Among the many foreseen uses for Unmanned Aircraft Systems (UASs), transportation is viewed with great interest. Several worldwide companies from delivery and e-commerce industries such as Amazon, FedEx, UPS, and DHL have begun trials using drones for

last-mile delivery, and academicians consider a potential for cost saving and improving responsiveness in comparison to traditional transportation modes [5,6]. In this comparison, however, drones are featured by a reduced capability in terms of raw amount of cargo weight that can be delivered. To offset this limitation, drone delivery can achieve profitability via transporting high-value cargo, reducing delivery time, and/or reaching hard-to-access areas.

All three of these can be leveraged in delivery operations of medical goods. In these operations, cargo value is not defined by its raw monetary value, but its perceived value in terms of need and potential cost savings, in this case related to complications due to belated treatments or even the loss of human lives. In addition, drones are not limited to ground transport infrastructure such as roads or airports, enabling them to heavily reduce delivery time and reach almost any delivery point. This is representative both for sparsely populated areas, where roads can be ill-conditioned or nonexistent, as well as densely populated ones, where traffic jams are a common issue.

Therefore, medical delivery operations are good candidates for the deployment of drone logistic networks. Indeed, in 2017 the delivery of medical drones was identified to have great potential for increasing system efficiency and saving more lives [7], enabling the provision of humanitarian aid in areas affected by natural disasters and emergencies with a more efficient response time, to reduce the delivery time of laboratory samples and products to remote health centers and hard-to-reach people, to perform organ transportation for transplantation, or even to supply automated external defibrillators (AEDs) to patients in cardiac arrest.

A methodical review of the current applications of drones in healthcare and health-related services shows that many of the published works discuss the technical theory behind network implementation [8] or system design for healthcare drone applications [9,10], proving that it is a market under development. Some of the currents research lines in medical drone delivery applications are focused on the transport of blood [11], organs [12], vaccines [13], AEDs [14–16], and medical samples [17–19].

A brief overview of the most relevant drone delivery actions is shown in [8]. Here, we can see the used schemes are not homogeneous, neither because of drone configuration (fixed wing versus rotorcraft) nor required ground infrastructure. For example, DHL and Matternet schemes rely on the deployment of ground stations to support operations. The cases where the cargo is delivered via parachute are limited due the delivery link not being bidirectional. In addition, these operations have been approved according to different criteria, depending on the specific drone regulation in each country. Thus, they do not represent a common base for the assessment of safety or the development of standardized operational procedures.

Seeking to harmonize drone regulation, the European Union (EU) has developed a common regulatory frame [20,21], which addresses operations in an operation-centric and risk-based approach. Following this approach, operations can be divided into three categories with increasing level of risk: open, specific, and certified. EASA is tasked with the development of associated guidance material (GM) and acceptable means of compliance (AMC) regarding these processes. From this point onward, we will use the terms unmanned aircraft (UA) and unmanned aircraft system (UAS) to refer to drones, as these are the terms adopted by EASA. The difference is that the term UAS includes ground control station (GCS) and any other equipment, while UA refers to the aircraft alone.

Inside the EU regulatory frame, delivery operations lie in specific category, as they need to be performed outside VLOS conditions to enable enough range for the benefits to be significative. Operations outside VLOS conditions are performed in either beyond visual line of sight (BVLOS) or in extended visual line of sight (EVLOS), the latter employing visual observers who must keep track of the drone. BVLOS operations, lacking visual contact with the drone, require the use of detect and avoid (DAA) systems to avoid collisions with obstacles or other aircraft.

Specific operations are subject to a risk assessment using a dedicated methodology, Specific Operations Risk Assessment (SORA). This risk assessment must represent a pre-defined, standardized Concept of Operations (ConOps) or alternatively be submitted for approval to the civil aviation authority of the corresponding country.

This paper aims to define and test a specific ConOps for UAS BVLOS medical delivery operations in urban areas and inside controlled airspace as an extra dimension to an existing ground logistics network, including the process of operational authorization. The ConOps definition features risk assessments using EASA-adopted SORA methodology, to assure representativeness of the process inside the upcoming EU regulatory frame. Test flights have been performed in three real scenarios, where UAS delivery is proposed as an alternative solution for ground-based transport. Due to the increased load caused by COVID-19, the proposed solution leverages the existing ground infrastructure to quickly deploy an air transport route, without the addition of specific infrastructure to support air operations. In addition, the possible needs for such ground infrastructure, personnel, and external infrastructure will be evaluated, assessing aspects such as operations' availability, scalability, and profitability to assess the maturity of the ConOps for large-scale implementation of medical delivery services.

## 2. Materials and Methods

First, it is important to note that UAS flights were performed prior to the entry of EU regulation, during 2020, and thus were subject to Spanish regulation. However, representativeness is maintained as Spanish UAS regulation requires an operational authorization under the scope of SORA, in the same way these operations would be performed inside EU regulation.

### 2.1. Logistics Network Model and Scenarios

In a logistics network, there is a flow of services and supplies with varying demand, with the purpose of the logistics distribution system being to deliver all supplies required to perform the demanded services. Both services and supplies can be classified in the following classes:

- **Basic**: can be directly input inside the network and consumed by a requester. Examples of these are generic treatments, food, protective gear, and preliminary or simple diagnosis.
- **Advanced**: obtained after the processing of basic services/supplies. Examples of these are difficult diagnosis, complex treatments or results from analysis, which require very specific equipment and/or trained personnel.

The proposed logistics distribution chain is hierarchized by nodes distributed in four levels:

1. **Distribution hubs**: they possess high storage capabilities and have a strong and reliable interface with other transport modes. These nodes act either as providers of basic supplies for the logistic network or as links between other nodes that lie too apart from the network.
2. **High-end medical facilities**: primarily hospitals or specialized testing centers. These nodes are featured by possessing major diagnosis, treatment, and/or analysis capabilities, being able to provide advanced services and supplies to the network. In contrast, these facilities have a high demand of basic supplies, needed to perform their duties.
3. **Low-end medical facilities**: these facilities have much lesser influence areas and diagnosis/treatment/analysis capabilities, sometimes entirely lacking any of them. For example, nursing homes and local health centers are found in this class. These facilities have a moderate need for basic supplies and will usually request advanced supplies and services, as they may not have the capability to procure those.

4.  **Non-specified requesters**: any general entity that requests services from the logistic network. Both planned (e.g., routine medicine delivery to rural or hard to access areas) and unplanned events (e.g., out of hospital cardiac arrest) are considered. The former will mainly require basic supplies and services, while the latter are featured by a need of advanced ones.

According to this classification of nodes and corresponding to current logistics needs in Valencia area, three test scenarios have been defined:

*   **Scenario A**: municipality of Siete Aguas. In a hilly area, an aerodrome ("La Loma") acts as a minor distribution hub, providing basic supplies to nearby rural and hard-to-access areas. The aerodrome is elevated 841 m above mean sea level (AMSL).
*   **Scenario B**: municipality of Gandia. A regional hospital (high-end facility) is connected to two low-end facilities: a nursing home (B-1) and a local healthcare clinic (B-2). The area is considerably flat, with no remarkable slopes. Terrain elevation near the hospital is 20 m AMSL and about 5 m AMSL in both the nursing home and the clinic.
*   **Scenario C**: municipality of Valencia. A major distribution hub is the Valencia Exhibition Center (Feria de Valencia), which has been temporarily repurposed as a logistics center due to COVID-19. This hub supplies hospital Arnau de Vilanova. There is a noticeable elevation difference between the logistics center (61 m AMSL) and the hospital (21 m AMSL).

Figure 1 shows the routes defined in scenarios B and C.

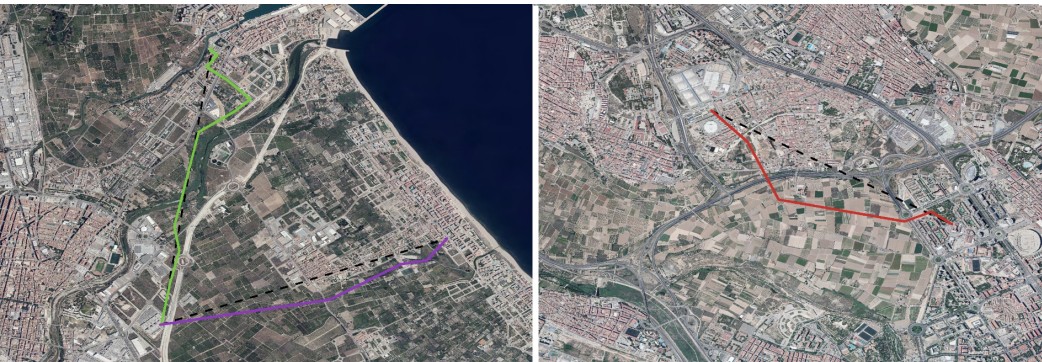

**Figure 1.** (**left**) Flight path for Gandia routes B-1 (purple) and B-2 (green). (**right**) Flight path for Valencia route C (red). Straight path is marked by the black discontinuous line.

### 2.2. Scenario ConOps and Risk Assessment

Under applicable Spanish law [22], urban UAS operations could be performed under an authorization, but were limited to VLOS conditions and a maximum take-off weight of 10 kg. In addition, transport operations were not allowed. To bypass these limitations, Article 43 of the work in [22] was applied. This allowed an exception for transport experimental flights over urban areas and outside VLOS conditions, due to the flights' purpose being the development of new techniques against the COVID-19 crisis. Flights were still required to comply with safety requirements such as equipping a parachute and being performed inside Temporary Segregated Airspace (TSA).

To obtain authorization for the operations by AESA (Spanish Safety Aviation Agency), a risk assessment using SORA methodology was performed for each of the scenarios. As SORA is currently under development, the version used as reference is the one adopted by EASA [23]. This version did not, at the time of performing the risk assessments, support BVLOS operations over urban areas, having EASA published a Notice of Proposed Amendment [24] to further discuss this matter. Therefore, option 1 of those contemplated in [24] was assumed for this project (i.e., use of original intrinsic Ground Risk Class (iGRC) values proposed by JARUS).

GRC evaluation requires the categorization of overflown ground areas between populated or sparsely populated areas, a process for which there is no quantitative literature at the time of performing the assessment. In this case, we followed EU indications based on ED Decision 2020/022/R, defining populated areas as those substantially used for residential, commercial, or recreational purposes. We considered these criteria to include street blocks, parks, streets, and urban roads, among others, and exclude fields or service roads. We defined exposure time as the total amount of time that the UAS would spend overflying the populated areas and defined the flight paths to minimize this value, at the cost of a slight increase in overall route length. Tables 1 and 2, respectively, show increase and decrease in length and exposure values of flight paths. Whenever overflight of populated areas could not be avoided, local police officers were deployed to avoid third-party entry, effectively turning these zones into controlled ground areas.

**Table 1.** Comparison between straight routes and low-risk paths' length.

| Scenario | Straight Route Length (m) | Low-Risk Route Length (m) | Length Increment (%) |
|---|---|---|---|
| B-1 | 2587 | 2663 | 2.94 |
| B-2 | 3023 | 3448 | 14.06 |
| C | 2384 | 2747 | 15.23 |

**Table 2.** Comparison between straight routes and low-risk paths' exposure (for 15 m/s cruise speed).

| Scenario | Straight Route Exposure (s) | Low-Risk Route Exposure (s) | Exposure Decreased to (%) |
|---|---|---|---|
| B-1 | 82 | 15.67 | 19.10 |
| B-2 | 50.2 | 33.6 | 66.93 |
| C | 84 | 18.47 | 21.99 |

This way, using SORA nomenclature, the scenarios are defined by operations conducted in BVLOS conditions over a mix of sparsely populated areas and controlled ground areas and inside "atypical airspace", corresponding to TSA. Although visual observers were required by AESA, being the operations effectively performed in EVLOS conditions, BVLOS conditions were simulated. In the scope of SORA ground risk scenarios, EVLOS is equivalent to BVLOS, and "BVLOS over a sparsely populated area" has higher iGRC value than "over controlled ground areas", so "BVLOS over a sparsely populated area" was considered as a reference scenario for iGRC.

Regarding ground risk mitigation, the following measures have been applied:

- **Strategic Mitigations (M1)**: applied with a low robustness level in scenarios A and B, and medium robustness level in scenario C, via people density assessment and coordination with local police corps.
- **Reduction of ground impact effects (M2)**: applied with a medium robustness level in scenarios B and C with the use of a parachute.
- **Emergency Response Plan (ERP) (M3)**: applied with a medium robustness level in all scenarios. The ERP is reinforced with the active involvement of local police corps and healthcare facilities.

The air risk section of the risk assessment is featured by the presence of atypical airspace, defined as the airspace with minimal encounter risk, as no other aircraft should enter the operational volume. In the case of Valencia, the TSA is considered as non-classified airspace, even it is within the Valencia Airport Aerodrome Transit Zone (ATZ). Notwithstanding, air risk mitigation measures equivalent to those corresponding to flight inside controlled airspace were applied, as well as DAA procedures and equipment. Regarding procedures, Filed Flight Plan (FPL) was submitted and approved, flight clearance was procured and VHF communication in tower-assigned frequency was maintained during the operation. With respect to equipment, the aircraft was fitted with Automatic Dependent

Surveillance-Broadcast (ADS-B) and a Mode-S transponder, as well as a forward-facing first-person view (FPV) camera.

Finally, based on ARC and final GRC (fGRC) levels, a SAIL value of II was obtained for scenario A and a value of I for scenarios B and C. SAIL stands for Specific Assurance and Integrity Levels and is an indicator of the overall level of confidence required by mitigation measures, directly proportional to operations' risk. Nonetheless, Operational Safety Objectives (OSO) were applied for a SAIL II value, as an extra safety layer. A summary of SORA parameters is shown in Table 3.

**Table 3.** Specific Operations Risk Assessment (SORA) parameters for scenarios. N = none, L = low, M = medium, * Mitigations for SAIL II were applied, ** Mitigations for flight in controlled airspace were applied.

| Scenario | Ground Scenario | iGRC | M1 | M2 | M3 | fGRC | Air Scenario | ARC | SAIL |
|----------|-----------------|------|----|----|----|------|--------------|-----|------|
| A | BVLOS spars. pop. | 4 | L | N | M | 3 | Atypical airspace | a | II |
| B | BVLOS spars. pop. | 4 | L | M | M | 2 | Atypical airspace | a | I(II) * |
| C | BVLOS spars. pop. | 4 | M | M | M | 1 | Atypical airspace ** | a | I(II) * |

### 2.3. Flight Procedures and Equipment

The flights were mainly performed with commercial off-the-shelf products, readily available in the market, as one of the main purposes is to evaluate the feasibility for a quick deployment of the network. This implied the use of open ISM (standing for Industrial, Scientific, and Medical use) radio bands for C2 link, which exhibit unreliable behavior in urban environments. As the control over the aircraft was one of the main safety requirements, a redundant scheme was deployed, with one pilot in each of the nodes. A summary of main flight parameters is shown in Table 4. In addition, these pilots were deployed in elevated places, to improve VLOS conditions and range. Flight plan consists in a piloted VLOS take-off and ascent to 50 m AGL, continued by an automated cruise phase following the planned route. Visual observers were deployed along the flight path, guaranteeing a maximum of 500 m horizontal distance between the aircraft and pilots or visual observers. Once cruise phase ends, the aircraft awaits the hand-over maneuver, in which control is manually transferred to the second pilot, who then lands the aircraft in VLOS conditions. Pilots and visual observers kept oral communication during the whole flight. In the event of control link loss, three different settings regarding fail-safe action were tested: continue the route, hover in position, and smart return to home (RTH) (i.e., backtracks the previously followed route instead of returning in a straight line). Flights were monitored from a control center, which received information regarding the location of aircraft and video signal from FPV camera. Flights required the involvement of two pilots, up to three observers, and around four to six local police officers, plus one overseer in the control center.

**Table 4.** Flight parameters.

| Take-off and Landing Mode | Cruise Phase Mode | Cruise Speed | Cruise AGL Height |
|---------------------------|-------------------|--------------|-------------------|
| Piloted VLOS | Automatic EVLOS/BVLOS | 15 m/s | 60–90 m |

Regarding materials, the following were used:

- **DJI Mavic 2 Enterprise**: this multirotor with a maximum take-off mass of 1.1 kg was employed for initial tests to assess the adequacy of the flight area and the level of radio interference. It was equipped with a **SafeAir Mavic** parachute manufactured by ParaZero. It was not used to transport any cargo.

- **DJI Matrice 300 RTK**: this was used for the transport tests. Its maximum take-off mass is 9 kg and supports hand-over procedure, which is a key feature for double pilot redundancy. It embedded a **FlySafe-M3** parachute manufactured by Flying Eye and a **Ping20Si** Mode-S ADS-B transponder, manufactured by uAvionix. Two DJI Smart Controller Enterprise were used as GCS, each of them fitted with a FlySafe-M3 manual activator. During flight #8, the aircraft was fitted with **DJI Zenmuse H20T** camera payload. According to the manufacturer, this UAS has a nominal range of 8 km (under CE conditions), without obstacles nor interferences.
- **Cargo bay for DJI Matrice 300 RTK**: we manufactured a prototype cargo bay for the flight tests. It consists of two separate pieces: a container and a base. The cargo bay base is fitted either over the top or under the bottom of the center part of the aircraft and is anchored using the same screw holes employed to fix the gimbals for the standard camera payloads (see Figure 2). The container can easily be attached to and detached from the base after and before flights using safety pins to lock it in place. Three different cargo bay container models with varying geometry were tested during flights (see Table 5 and Figure 3). These were designed not to interfere with obstacle-sensing cameras on the top and sides of the aircraft so that obstacle avoidance capabilities are not impaired. In addition, locating the cargo in the same spots intended for standard payload cameras reduces the occurrence chance of issues related to gravity center displacement and balance.
- **DJI Flight Hub**: software used as control center. It was installed in a web server based in Paris (Europe).
- **Portable 4G LTE modems**: these were carried by the pilots and used to provide the link between the GCS and the control center for data transmission.

**Table 5.** Base and container parameters.

| Name | Weight (g) | Size (cm) | Description |
|------|-----------|-----------|-------------|
| Base-down | 190.62 | 14 × 18 × 4 | Base anchored in the lower side |
| Base-up | 203.8 | 14 × 18 × 4 | Base anchored in the upper side |
| P1 | 61.8 | 10 × 15 × 8 | Paralelepiped, fitted for vial transport |
| P2 | 67.4 | 10 × 15 × 18 | Taller than P1, with sloped back. Fitted for PPE |
| P3 | 296.85 | 10 × 12 (radius, height) | Dome-shaped and robust. Higher capacity |

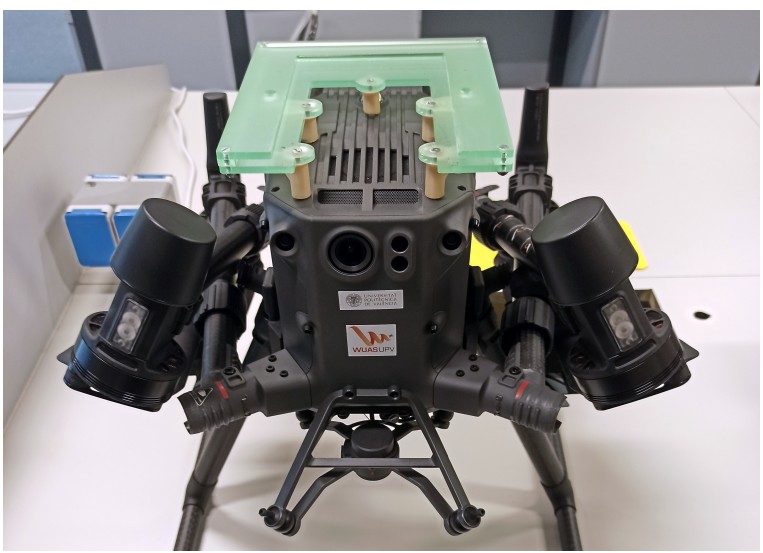

**Figure 2.** DJI Matrice 300 RTK fitted with upper cargo bay.

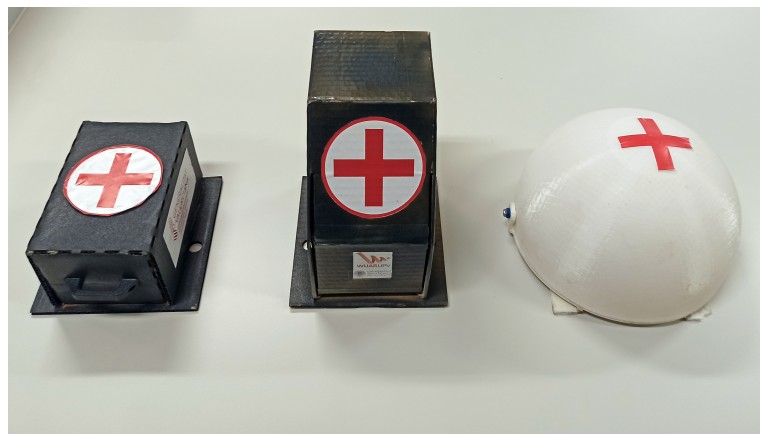

**Figure 3.** Containers P1 (left), P2 (center), and P3 (right).

The goods chosen for the performed tests were protective personal equipment (PPE) and COVID-19 test units, as well as vials simulating medical samples. Table 6 collects the weight of different PPE items and test units. Maximum cargo capacity, considering the cargo bay and the payload, is 2 kg for the tested UA. As many of the items cited in Table 6 feature a very low density, the limitation was not the maximum weight but the total volume of the cargo. The total weight of the UA during the tests did not exceed 8 kg.

**Table 6.** Weight of transported medical goods.

| Item | Weight (g) |
|---|---|
| Safety googles | 78.5 |
| FFP2 mask | 6.7 |
| Gloves (×2) | 6.6 |
| Protective full suit | 120.4 |
| COVID-19 test | 6.6 |
| Sample vials | 3–30 |

## 3. Results

A series of test flights was performed in the three defined scenarios, during October and November 2020. Results from flights in scenario A will only be briefly commented in this paper, as these were preliminary tests and operations in rural areas and uncontrolled airspace have been already assessed by several other authors [11,14]. These flights were performed on days 19 and 31 October 2020. During them, a time-out error was occasionally shown by the GCS when performing the hand-over procedure. To cope with this error, a forced hand-over procedure was developed (i.e., manually switching off the in-control GCS). Table 7 summarizes the main parameters of performed flight tests in scenarios B and C. Note that this table only collects EVLOS/BVLOS flights performed with DJI Matrice 300. Preliminary VLOS flights for obstacle and performance evaluation will not be addressed.

On the one hand, flights performed in scenario B sustained no remarkable event that could affect the flight mission or safety. Pilot 1 was positioned in a vantage position (e.g., hospital rooftop), which granted direct visual contact with the aircraft up to 2 km away and a good C2 link quality throughout most of the flight. Hand-over procedure required forced activation due to time-out.

On the other hand, flights performed in scenario C, which are more complex than B, were subject to a varied number of safety-threatening events. During the first flight in this scenario (#3), cruise height topped 60 m AGL as a compromise between ground and air safety margins. Considering that the elevation difference between departure and arrival points is 40 m, the result from pilot 1 perspective is a descending trajectory where, at the end, the aircraft is only 20 m above the pilot. In addition, GCS 1 experienced heavy interference at pilot 1 initially planned position, which was updated ad hoc. The coupled

effect of both situations caused the flight path to be obstructed by nearby structures, lowering pilot 1 situational awareness and C2 link quality. Eventually, GCS 1 link was lost at around 800 m from pilot 1, much earlier than expected, taking pilot 2 aircraft control. As GCS 2 signal was weak, it was decided to abort the flight: pilot 2 maneuvered the aircraft outside of the interference area so that pilot 1 could take back control and land in the starting position. Cruise AGL height was modified to match 60 m over the logistics hub, increasing up to 90 m over the hospital, to compensate for the elevation difference.

**Table 7.** Performed flights.

| No. | Scenario | Date | Departure | Arrival | Height | Payload | Event |
|---|---|---|---|---|---|---|---|
| 1 | B-2 | 02/11 | Hospital | Local clinic | 60 m | P1 | none |
| 2 | B-2 | 02/11 | Local clinic | Hospital | 60 m | P1 | none |
| 3 | C | 03/11 | Logistics hub | Hospital | 60 m | P1 | Low flight height |
| 4 | C | 09/11 | Logistics hub | Hospital | 60–90 m | P1 | none |
| 5 | C | 09/11 | Hospital | Logistics hub | 60–90 m | P1 | none |
| 6 | C | 10/11 | Logistics hub | Hospital | 60–90 m | P2 | ATC coord. failure |
| 7 | C | 10/11 | Hospital | Logistics hub | 60–90 m | P2 | ATC coord. failure |
| 8 | C | 10/11 | Logistics hub | Hospital | 60–90 m | P2 | GCS 2 unsynced |
| 9 | C | 13/11 | Logistics hub | Hospital | 60–90 m | P2 | C2 loss |
| 10 | C | 13/11 | Hospital | Logistics hub | 60–90 m | P2 | Ballon flight path |
| 11 | C | 13/11 | Logistics hub | Hospital | 60–90 m | P2 | none |
| 12 | C | 13/11 | Hospital | Logistics hub | 60–90 m | P2 | none |
| 13 | C | 16/11 | Logistics hub | Hospital | 60–90 m | P3 | none |
| 14 | C | 16/11 | Hospital | Logistics hub | 60–90 m | P3 | none |
| 15 | B-1 | 16/11 | Hospital | Nursing home | 80 m | P1 | none |
| 16 | B-1 | 16/11 | Nursing home | Hospital | 80 m | P1 | none |

Operations were resumed on day 9 after the approval of both the updated path and higher height TSA. Flights #4 and #5 featured an acceptable C2 link quality and permanent visual contact between pilots and aircraft. However, FPV camera image quality experienced constant drops for both GCS. This issue persisted in all flights, even at moments when the aircraft was close to the pilot. Although flights #6 and #7 did not feature any safety event during the operations, subsequent communication with ATC showed a lack of effective coordination between both parties: operator and air navigation service provider (ANSP). As flights were performed inside TSA, which is non-classified airspace, ANSP's operations control department deemed FPL submission as unnecessary. This information was passed to the operator, but not to local ATC office. This office, noticing a lack of FPL for flights #6 and #7, contacted the operator and demanded FPL submission for subsequent flights. In addition, ATC informed the operator that the aircraft's transponder signal was not received. Both parameters were checked, and flights #8 to #14 featured FPL and working transponder signal, which was confirmed by ATC.

To cope with weak FPV signal, we decided to test the DJI Zenmuse H20T payload as an alternative means of on-board camera. Results were positive in VLOS conditions, where it provided better quality than the FPV camera. However, the UAS supplier alerted the operator that the H20T payload uses an extensive amount of the aircraft's computational and bandwidth capabilities, which could hinder long-range operations. Flight #8, in which the aircraft was equipped with H20T, was aborted after completing around 75% of cruise phase, because GCS 2 unlinked from the aircraft and control could not be reestablished during flight. The aircraft lost C2 link with pilot 1 at that point, entering fail-safe hover mode. Pilot 1 had to reposition to a higher ground, where control was recovered, bringing the aircraft back home in manual mode. Therefore, it was decided to perform further tests with H20T later, before embedding it again for EVLOS/BVLOS flights. After the experience from the last flight, during flight #9 fail-safe action was set to smart RTH, which would retrace the flight path back to the starting point. At one moment during flight #9, both GCSs lost control simultaneously near 70% of the route, prompting fail-safe smart

RTH mode. However, GCS 2 quickly recovered control link and pilot 2 aborted RTH and navigated the UAS to arrival point, following the planned route.

Flight #10 featured an unexpected event after an intruder broke into the TSA and the flight path. After finishing take-off and ascent, just after starting automatic route, a heart-shaped, helium-filled balloon with a hanging note was spotted crossing the vicinity, very close to the route path. The pilot decided to halt the automated flight, keeping the aircraft in hover position until the intruder had crossed the operation volume. No other safety-related event happened during the rest of flights (#11 to #16), apart from the low FPV quality and occasional forced hand-over activation. C2 link exhibited unreliable behavior once the distance between the aircraft and the GCS was higher than 2 km. In addition, C2 link could not be established between the UAS and the further GCS during take-off and landing maneuvers, until the aircraft reached around 40 m height. No manned aircraft entered the vicinity of the operation volume, neither any other UAS was spotted during any of the flights. During flights #3, #4, and #5, the transponder was fitted inside the container, instead of loading medical supplies. The rest of flights were performed transporting a variety of PPE equipment and COVID-19 test units.

ATM coordination, although in the end successful, was remarkably laborious, with the involvement of two different ANSPs (ENAIRE and FerroNATS) and the airport manager entity (AENA). After FPL submission, a phone call was usually required, as ANSP detected inconsistencies in the provided information due to an unfit submission form. An example of this was caused by declared endurance versus flight time. Manned aircraft have a higher endurance than flight time, but in the case of UAS, endurance is lower, and several battery sets are used.

Coordination with local police corps, in charge of securing ground-controlled areas, was seamless, with fluent communication and deployment. The need to coordinate with local police corps, however, reduced the overall availability of operations, due to additional constraints. Some hospital workers, who were unaware of this project, showed enthusiasm regarding the use of UAS for transport of medical goods.

## 4. Discussion

In the upcoming EU regulatory frame, BVLOS transport operations lie inside a specific category and will be subject to a SORA assessment prior to their approval. This approval shall be emitted after an authorization application, following a procedure very similar to the one in this project. There is a subtle exception for these specific category flights by which, when transporting "dangerous goods", which must be stored inside a container that would prevent leakage and third-party exposure in the event of an accident. Many medical goods are labeled as dangerous, such as pathogen samples or even untested blood, so certain precautions will need to be taken when transporting them.

15th December 2020, EASA published a decision regarding [24] proposals for SORA GRC values (ED Decision 2020/022/R) [25]. The adopted option is the same that was assumed during the performed risk assessments. Therefore, their representativeness regarding upcoming EU regulation is maintained regarding this aspect.

It is important to note that, as exposed in Table 3, several additional layers of mitigation measures were implemented as extra safety layers. In addition, the lack of dedicated ground and external infrastructure was also a limiting factor. Another aspect to consider is that flights were not performed inside an urban center but in peri-urban and inter-urban scenarios where most of the flight path could be redirected towards sparsely populated areas. According to Table 2, benefits of route planning differ greatly depending on the specific urban layout in the operational area. Because of the mentioned reasons, even if operations were deemed feasible, they were subject to several restraints that hamper their performance in terms of availability, scalability, and profitability.

Operations availability, in terms of area coverage, is directly linked to their maximum range. Reasons why actual operating range (i.e., C2 link range) was reduced, compared to nominal values, are assumed to be related to the level of electromagnetic interference.

Primarily, due to the use of already crowded ISM bands and, by a minor degree, because of nearby metallic structures, which cannot always be avoided in urban environments. Although the use of cellular networks can serve to bypass these limitations, those should comply with a minimum of performance requirements, as an external system supporting the operation, in which security is also one of the key aspects. We believe a robust solution is the use of both technologies: a redundant scheme where a ubiquitous and reliable cellular network serves as a primary C2 and data link, while point-to-point RF connection in a UAS-specific frequency band acts as back-up. This RF link can be achieved via emitting from dedicated ground infrastructure (e.g., vertiports), using relays when deemed necessary.

Another aspect directly related to availability is meteorological conditions, which can affect both the aircraft and the pilot, if exposed. Valencia's climate is dry and relatively hot, its terrain is mostly flat, and wind intensity is usually low. Thus, no availability issues were derived from meteorology during this project. Other locations, however, may be subject to frequent rains and/or strong winds, which may severely affect the availability of operations. On the one hand, the use of cellular networks or dedicated ground infrastructure shall erase the need to deploy an exposed pilot. On the other hand, SORA methodology includes mitigation measures related to meteorology, both considering operator procedures and aircraft performances. These measures shall be used as a reference in cases where adverse meteorological conditions are expected.

One of the main factors limiting the scalability and profitability of operations the way we performed them is the relatively high amount of involved personnel needed (up to 10 people). This aspect is also highlighted by Leonardo in their recent experience after performing flights in a very similar scenario [26,27]. To enable a higher scalability and profitability of operations, the amount of involved people must be reduced. The following ways illustrate examples regarding how this can be achieved.

- Operating over controlled ground areas allows a moderate reduction of SORA GRC value, at the cost of having to secure said areas, which usually implies personnel deployment and the requirement to use fixed routes. Withdrawal of controlled ground areas raises the intrinsic GRC level to a value of 6, corresponding to "BVLOS over populated areas" category. The use of all three ground mitigation measures with a medium level of robustness allows for a final GRC of 3, meaning SORA compliance can still be achieved without an increase in SAIL. However, this would still limit the operations availability to routes and timeframes where/when less people are exposed compared to nominal times. To avoid these limitations, ground strategic mitigations (M1) can also be withdrawn. In this case, the higher exposure shall be compensated with the enhancement of overall safety via compliance with requirements linked to higher-than-II SAIL levels and their linked Operational Safety Objectives (OSOs). Regarding this matter, 17th December 2020 EASA published a Special Condition developing UAS certification requirements for operations in SAIL III and IV.
- Operating in the EVLOS condition requires the deployment of visual observers. In this case, visual observers were used as an additional mitigation measure, so they can be removed without losing SORA compliance.
- Finally, pilots can also be removed from the operation site. For this purpose, employing a centralized control center is a feasible solution. Operations performed this way can have varying degrees of automation, up to the point of only requiring active control during specific abnormal or emergency situations or even not active control at all.

When applying these measures, it should be noticed that situational awareness must still be guaranteed via other means. We can take flight #10 as an example, where enhanced control and situational awareness via deployment of pilots and visual observers allowed the detection of a balloon entering the flight path and the timely execution of contingency procedures. This event, although unlikely, is just one example of the many unforeseen situations an unmanned aircraft can face during urban operations. Integrating this kind of operation inside a populated ground and airspace requires fully operative navigation

and DAA capabilities. For these purposes, the deployment of UTM/U-Space and their associated services will be a key enabler of large-scale operations. Regarding integration between UAS and ATM, our experience supports the need for specific UTM procedures, as traditional ATM is not fit for UAS operations. This led to sub-optimal coordination, creating a higher-than-needed workload for both parties (operator and ANSP), which limits the number of operations that can be supported. Future developments shall support a simplified interface between operators and ANSPs.

## 5. Conclusions

Considering that the relatively small number of performed flights does not serve as a basis to guarantee the level of safety of this kind of operations were they performed at a large-scale level, this project has served to prove the feasibility of the UAS delivery ConOps, which can be quickly deployed as an extra support for ground transport in dire situations, without the need for dedicated infrastructure nor external services supporting the operation, other than the base ground logistics network infrastructure.

However, the proposed ConOps has several limitations in its current state, not having reached the required level of maturity for large-scale deployment. Advancements still need to be achieved, especially regarding support for BVLOS operations and integration within a populated airspace. This project serves as an experimental use case which shall feed the development of U-Space/UTM services, certification standards, and regulatory procedures that will enable large-scale deployment of UAS operations.

**Author Contributions:** Conceptualization, I.Q.G., N.V.V., and J.V.U.; methodology, I.Q.G., N.V.V., and J.V.U.; software, P.A.M.; validation, N.V.V. and J.V.U.; formal analysis, N.V.V. and J.V.U.; investigation, N.V.V., P.A.M., and J.V.U.; resources, J.V.U. and B.F.G.; data curation, N.V.V. and J.V.U.; writing—original draft preparation, N.V.V. and P.A.M.; writing—review and editing, N.V.V. and I.Q.G.; visualization, J.V.U. and B.F.G.; supervision, I.Q.G.; project administration, I.Q.G.; funding acquisition, I.Q.G. All authors have read and agreed to the published version of the manuscript.

**Funding:** This research was funded by Generalitat Valenciana (DECRETO 63/2020) and Universitat Politècnica de València (PAID-01-18). The APC was funded by Universitat Politècnica de València.

**Institutional Review Board Statement:** Not applicable.

**Informed Consent Statement:** Not applicable.

**Acknowledgments:** We would like to thank Valencia and Gandia local police corps for their involvement in route planning and area control as well as Valencian Emergency and Safety Response Agency, AVSRE, for lending the use of La Loma Aerodrome. Assistance provided by healthcare facilities was greatly appreciated (Feria de Valencia, Hospital Arnau de Vilanova, Hospital Comarcal Francesc de Borja, Residencia Solimar Daimús and Centro de Salud del Grau). We also express gratitude towards ENAIRE, FerroNATS and AENA for their role in coordination operations inside an ATM structure.

**Conflicts of Interest:** The authors declare no conflicts of interest. The funders had no role in the design of the study; in the collection, analyses, or interpretation of data; in the writing of the manuscript; or in the decision to publish the results.

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
