# Peer review of "A Quickly Deployed and UAS-Based Logistics Network for Delivery of Critical Medical Goods during Healthcare System Stress Periods: A Real Use Case in Valencia (Spain)"

_drones, doi:10.3390/drones5010013_

Round 1
Reviewer 1 Report
This is a timely addition to the growing field of drone healthcare delivery. The more test cases available for others to to draw on in planning similar experiments the better. Due to the challenges (both regulatory and technical) of operating in developed countries and in built up areas such a case study in Europe is a valuable example.
I do have some points that I hope the authors find helpful:
- The authors state that the aim of the paper is to "test the feasibility of an UAS logistics network for medical delivery, without relying on ground infrastructure or major external systems supporting the operation", yet it appears that Scenario's A and C do operate from a distribution point where applicable infrastructure is in place. It might help to clarify that no additional infrastructure specifically for the operation of the drone was required.
- In section 2.2 it appears special exemptions were obtained to conduct experimental flights, but I wonder how this translates into real world potential if a permanent UAS for healthcare delivery was to be proposed. The presence of police officers and a change in status of the air and ground zones could almost be interpreted as introducing structure.
- Regarding the equipment - why were DJI drones chosen exclusively and what was the reason for choosing the DJI Matrice 300 RTK? Could security be considered an issue, especially given that DJI has been sanctioned recently?
- Figure 2 - it would have been appropriate to have a second photo of the cargo bay and payload capacity for visual reference (scale).
- It would be interesting to briefly add in the acceptance and perceptions of the people outside of the project team (police, hospital staff...).
- The discussion rightly spends time on the issues of scalability and the need for some type of infrastructure to support the mission. The experimental flights also appear to have undertaken in good weather conditions, what might poor conditions mean for continuity of a critical healthcare drone service (and the pilot having to operate on high ground in the open!)? In addition, flight paths are unlikely to be fixed in a real and ongoing operational scenario, so drone regulations would need to be centrally revised. How will the new EU regulations impact healthcare focused drone operations? If drone logistics services take off then the skies are only going to become more crowded...
- One of the main questions that I do not think was answered in the discussion was whether this type of service will actually add value to existing healthcare delivery at the test site. Do the authors envisage a network of health service drones in Spain or is this purely an experimental urban test ground?
- Covid-19 is dropped into the abstract yet does not feature in the paper, not at all in the discussion.
I am not not an expert on local Spanish drone regulations so cannot comment on this aspect of the experiment, however, it appears great efforts were made to comply with the directives.
Lastly, there are issues with English language that must be tightened (spelling, grammar etc - too many to list in these comments). I believe this just needs careful copyediting.
Overall, on a positive note the paper highlights many of the hurdles impacting the wider deployment of UAS for healthcare delivery. Scalability will only come from conducting these types of test cases and it is important to report field experiences both positive and negative if progress is to be made.
Author Response
Thanks for your review, it has been very helpful. I will answer your comments in a numerated list:
- Regarding this matter, the text already included “dedicated infrastructure nor external services supporting the operation”, in the first paragraph of section 4. However, to clarify, we have added “other than the base ground logistics network infrastructure”. Also, we added a reference in the modified last paragraph of the introduction.
- Regarding obtained exemption and potential for permanent delivery, the text included references that this kind of operations will require a very similar approach to obtain permission for flights (non-experimental). However, they were somewhat vague. We have extended the second paragraph of section 4 indicating that BVLOS transport operations lie in specific category inside EU regulatory frame, and as such are subject to an operational approval, which shall follow a similar procedure to the one followed in this project. Regarding police and air/ground zones status, section 4 already discusses these aspects (bulleted list and following paragraph), suggesting their removal and substitution for alternate safety measures.
- In our operator, we have been using mostly DJI drones for 4 years, thus we are inclined to use DJI because of our familiarity with this brand. Nonetheless, DJI matrice 300 RTK was chosen for the BVLOS tests because it allows to perform a hand-over procedure, transferring control between two pilots. This is one of the main requirements of our approach, and not many commercial drones have that feature. We have added a brief reference in 2.3 to this. In addition, urban flights in Spain are limited to 10 kg of mass, and Matrice 300, having 9 kg, allows for a reasonable cargo capacity without surpassing the 10 kg limit (since we were operating under exemptions, maybe an exemption regarding maximum mass could have been emitted, but it would likely have further complicated things). We have added a reference indicating that Spanish regulation did not allow more than 10 kg in the first paragraph of section 2.2. Regarding security matters, that lies outside the scope of this project. However, we can say that the version of DJI Flight Hub we used directly sent the data to a server in Paris. We have added a brief reference to that aspect in section 2.3.
- We have added one more image with all three cargo bays and better resolution.
- In fact, some hospital workers, who were not aware of our project, seemed enthusiastic about it, as they saw us wearing pilot equipment (bright orange jackets, control radios, communication, etc.). During the landing phase of the first BVLOS flight, some hospital workers even opened their offices’ windows and cheered at the approaching drone. We were not sure about adding this aspect to the manuscript, as it is somewhat non-rigorous, but now we have added two lines at the end of section three reflecting this.
- Very nice observation. During the starting authorization phase of the project, historical meteorological conditions in the area (mainly wind and rain) were evaluated to check for a high level of availability. As Valencia’s climate is relatively dry and hot, with generally low levels of wind (mostly except in case of maritime event), we had no availability issues and thus did not consider this aspect inside the manuscript. We have added a paragraph in section 4 where we discuss aspects related to weather conditions and availability. The requirement of having fixed routes is given, in this case, due to the use of ground-controlled areas as mitigation measure, and, to a minor extent, due to the use of strategi mitigations. The first point in the bulleted list in section 4 describes this issue. However, we have detected that we did not specify that fixed routes were a consequence of these measures, and we have updated this paragraph for this matter. Finally, regarding EU regulations in a crowded airspace, this is mentioned in the paragraph after the bullet list: the UTM/U-Space services are expected to cover these aspects. In the case of Europe, regulation regarding U-Space is currently under development and as such we cannot include any statement regarding that matter.
- Following the indications of other reviewers, we have added a brief conclusion, where this aspect is treated.
- One of the goals of this manuscript is to try to generalize a “high stress” condition on healthcare systems, such as the one caused by COVID-19. We thought that results would lose value after COVID crisis ends if we focused too much on it. We have updated the last paragraph of the introduction, which now mentions COVID-19 and also better delineates the scope and objectives of the article.
We have also performed style corrections, along with modifications proposed by the other reviewers.
Reviewer 2 Report
The authors have presented a feasibility study of deploying unmanned aerial system (UAS) for delivering medical goods and supplies to the areas with limited access due to the lack of infrastructure. The authors have done a detailed risk assessment study following the standard guidelines provided by the appropriate authority. Finally, they have provided some recommendations that could make UAS a cost-effective and feasible option.
While this is a very important application area for UAS, the authors could have done a better job in presenting their study that could benefit others interested in deploying the technology in similar conditions.
The reviewer has following comments:
1) The authors should clearly state the scientific contribution of their study. They should clearly define the scope of the study. For example, did they incorporate the effect of weather pattern (wind, fog, rain, cloud, etc.) to their study?
2) When making recommendation, the authors should also comment on hardware, payload etc. What can be improved in the technological front? What is the limitation of current technology that might be proved as a roadblock for widespread deployment? That should serve as a guideline for the readers who might be interested in deploying UAS for delivering products other than medical supplies.
3)"Results from flights in scenario A will only be briefly commented in this paper, as these were preliminary tests and operations in rural areas and uncontrolled airspace have been already assessed by several other authors" - needs a reference pointing to the study by "other authors"
4)"For these purposes, the deployment of UTM/U-Space and their associated services [ref]…." please provide the reference.
5) "This aspect is also highlighted by Leonardo in their recent experience [ref]…" please provide the reference.
6)"Three different cargo bay container models with varying geometry were tested during flights (see Table 5 and Figure X)" - which Figure X?
7) "Withdrawal of controlled ground areas, according to [decision].." What is meant by "[decision]"?
8) The manuscript should be thoroughly checked for grammatical mistakes and typos. Some obvious ones noticed by the reviewer:
-Some of the currents research lines ….
-The cases where the cargo is delivered via parachute, are limited due the delivery link not being bidirectional
-This paper aims to test the feasibility of an UAS logistics…..
Author Response
Thanks for your review, it has been very helpful. I will answer your comments in a numerated list:
- We have updated the last paragraph of the introduction, to clarify the scope and objectives of the project.
- Regarding a comment on hardware, payload, and technology, especially those which could be a roadblock, the manuscript already comments in the discussion that a solution to actual “RF+one on-site pilot+ground controller” paradigm could be the use of cellular networks or dedicated infrastructures such as vertiports. We also briefly address the need to use fully operative navigation and detect&avoid capabilities as enablers of integration inside populated air and ground spaces. As the research is focused on the operational concept and requirement, we believe this comment is already enough regarding this matter.
- Missing references added
- Missing references added
- Missing references added
- Figure references corrected
- Reference to [decision] was a mistake, paragraph updated.
- We have checked grammar and typos.
We have also performed style corrections, along with modifications proposed by the other reviewers.
Reviewer 3 Report
This manuscript presents a logistics network model for the distribution of medical supplies via UAS. Three study sites, representing three realistic use-case scenarios, are presented and subjected to Specific Operations Risk Assessment (SORA) analysis to determine the feasibility of the aforementioned UAS logistics network. At least 16 test flights were performed across the three study sites to demonstrate the feasibility of UAS delivery operations within each use-case scenario.
While the study is structured and performed well, the presentation needs attention.
1. Reading this manuscript is difficult because of the extensive use of acronyms, many of which are not properly introduced (while some are not introduced at all). The first mention of an acronym must be accompanied by its meaning, both in the abstract and in the main text. I would also strongly recommend making an attempt to reduce the number of acronyms used overall, but I understand that is not always possible, especially when writing about government regulations. (Does anyone love acronyms more than a government agency?)
2. Extensive proofreading is required. I cannot provide a copy of my proofreading notes; I had to stop highlighting errors early on because proofreading was distracting me from analyzing the substance of the manuscript.
3. The final paragraph of the Introduction does briefly outline the study, but it is perhaps too brief. I suggest another one or two sentences to better outline the study. This would have helped prepare me as the reader to continue reading into the Materials and Methods section. You could also include a brief Conclusion section, no longer than two paragraphs, that succinctly states what the study has accomplished.
4. This is only a suggestion that you may not find to be useful, but: I am surprised at the low number of figures in a study such as this one. One figure I was hoping to see but did not was some sort of flowchart of how your applied SORA to your flight areas, and how you made your decisions altering those flight paths to mitigate risk. If this is something you could create in a short amount of time, I believe it would be beneficial.
This study is impressive in that it simulates a UAS delivery logistics network using only commercial, off-the-shelf products and no dedicated infrastructure, all while adhering to VLOS restriction. You justify well the contingencies that took place during the flight tests, and you wisely passed over presenting the results from Scenario A that were not pertinent to the study's aims of demonstrating the feasibility of your presented network model.
(I am also quite happy to see that you did not destroy what must have been a heartfelt note attached to the balloon during Flight #10. Perhaps your next study could propose a UAS logistics network for delivering love letters.)
My recommendation is to accept this manuscript minor revisions, as my only concerns with the manuscript are in the presentation of the study, and not the study itself.
Author Response
Thanks for your review, it has been very helpful. I will answer your comments in a numerated list:
- We have reviewed the use of acronyms in the manuscript. We have erased some of them (like VO, MTOM, OCA, RA) and substituted them by the worded expression. We have also properly introduced the terms, especially in the abstract.
- We have performed style correction in the manuscript.
- We have updated the last paragraph of the introduction, which now defines the objectives more clearly. In addition, we have also added a brief conclusion.
- Indeed the number of figures is low. Since SORA application process is already exposed in SORA and EASA documents (Here, page 41 https://www.easa.europa.eu/sites/default/files/dfu/Easy%20Access%20Rules%20for%20Unmanned%20Aircraft%20Systems%20%28Regulations%20%28EU%29%202019-947%20and%20%28EU%29%202019-945%29.pdf) and we followed that very process, we did not feel it was necessary to include a flowchart. Sadly, we do not have enough time to develop any additional graphic, but we have been able to update the pictures of the drone and payload, with more quality.
We have also performed style corrections, along with modifications proposed by the other reviewers.
Reviewer 4 Report
This is a very interesting article by way of being an in-depth empirical contribution to challenges facing the new branch (UAS etc.) within aviation per se. Good actualisation related to the potential for UAS with regard to pressure on the health care system due to today's pandemic, but the actualisation is also valid beyond situations under pressure, i.e., ordinary operations in normal times. The article demonstrates precise knowledge related to the aviation industry. The article also situates itself (and drones per se) within the traditional aviation industry by focusing on safety issues including mitigation of risks, and by avoiding a general approach to describing own methods, which is very positive. The article is generally very well written and to the point. Obviously, any discussion part has potential to dwell even more with some of its aspects. However, the article acknowledges challenges related to coordination with ATC together with current ATM not being fit for UAS operations. Also, the article mentions automation per se, which makes the future of control rooms relevant. These aspects highlight a multidisciplinary approach, thus some reflections on how technology, humans and organisation can work together (based on the empirical material) with regards to future UAS would have been interesting. Anyway, I believe the article needs to state even more clearly its research aim, and to pursue this even more explicitly in the discussion. In summary, the view is that this is a well worded article that indeed contributes to increased knowledge on UAS usage. However, there are some points that I would suggest looking into and clarify:
- Some more elaboration needed on the rationale behind the three scenarios developed and the parameters included. Are there links to aspects identified in the research literature etc.
- Clarify scope of article, perhaps via distinct research question(s). Safety vs. efficiency etc.
- Clarify findings from the results section, for example in tabular form.
- The discussion mentions "noticeable reduction in time and monetary costs". It would be nice to know what the basis for comparison is.
- Preferably an explicit conclusion where the article's new academic contribution is clarified
Author Response
Thanks for your review, it has been very helpful.
- We have updated the last paragraph of the introduction to clarify the scope and objectives of the manuscript. In addition, we now believe this is more explicitly related to what is discussed in section 4.
- We have also created an explicit conclusion, where we have moved the first paragraph of the discussion (we have erased the reference to the noticeable reduction in time and costs as we have no explicit reference).
We have also performed style corrections, along with modifications proposed by the other reviewers.
Round 2
Reviewer 2 Report
The authors have addressed the concerns of the reviewer.